# THINKSUM: PROBABILISTIC REASONING OVER SETS USING LARGE LANGUAGE MODELS

## ABSTRACT

Large language models (LLMs) have a substantial capacity for high-level analogical reasoning: reproducing patterns in linear text that occur in their training data (zero-shot evaluation) or in the provided context (few-shot in-context learning). However, recent studies show that even the largest LLMs fail in scenarios that require reasoning over multiple objects or facts or making sequences of logical deductions. We propose a two-stage probabilistic inference paradigm, THINKSUM, that reasons over sets of objects or facts in a structured manner. In the first stage (THINK – 'fast' retrieval of associations), a LLM is queried in parallel over a set of phrases extracted from the prompt or an auxiliary model call. In the second stage (SUM – 'slow' probabilistic inference or reasoning), the results of these queries are aggregated to make the final prediction. We demonstrate the advantages of THINKSUM on the BIG-bench suite of evaluation tasks, achieving improvements over the state of the art using GPT-family models on ten difficult tasks, often with far smaller model variants. We compare and contrast THINKSUM with other proposed modifications to direct prompting of LLMs, such as variants of chain-of-thought prompting. We argue that because the probabilistic inference in THINKSUM is performed outside of calls to the LLM, THINKSUM is less sensitive to prompt design, yields more interpretable predictions, and can be flexibly combined with latent variable models to extract structured knowledge from LLMs.

## 1 INTRODUCTION

Large language models (LLMs) (Brown et al., 2020; Rae et al., 2021; Chowdhery et al., 2022) can recall a broad range of basic facts, recognize and mimic various forms in language, and efficiently extrapolate analogies in structure and meaning. These abilities allow LLMs to excel in zero-shot and few-shot tasks that are formulated as generation or selection of a likely completion of a prompt. This formulation requires LLMs to perform associative **fast thinking**, in which each token of text in the sequence making up the answer is generated or scored in one pass through the model and, other than that, no intermediate information is created or retained. Fast thinking is made possible by the compression in the LLM weights of information that is repeated in a variety of ways in large training datasets.

However, it is increasingly evident that when **reasoning**, or **slow thinking**, is required, failure modes of LLMs are revealed. In our usage, reasoning is sequential manipulation of concepts that can be expressed in language. Tasks that require iterative retrieval of rarely stated knowledge, uncertainties over multiple objects or facts, or multiple steps of deduction are difficult even for the most advanced LLMs. In a recently designed suite of evaluations, BIG-bench (Srivastava et al., 2022), some of the tasks where the gap between machine and human performance is large involve inference sequences with nested counterfactuals (LOGICAL DEDUCTION), concepts introduced though definitions (CONCEPTUAL COMBINATIONS), etc. (see Fig. A.1). These are tasks where a human solver's intuitive feeling of '(in)coherence' is not sufficient to produce the right answer: the solution is obtained by a sequence of thoughts that can be explained in words and may even require writing down intermediate results if working memory is insufficient.

We show on several examples in BIG-bench that such problems can be addressed by a two-component mechanism, which we name THINKSUM:

- THINK (fast thinking / association / knowledge retrieval step): creating an association of spans of text with sets of strings. This process may involve generation from a language model, as is the case in Fig. 1, where the novel word 'binne' is associated with the set of strings {'cat', 'mink', . . . }

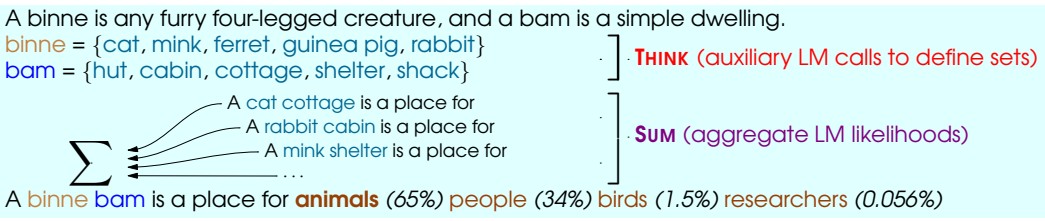

Figure 1: An example adapted from the CONCEPTUAL COMBINATIONS (INVENTED WORDS) task, in which models must select the most likely completion of a phrase that includes nonce words whose definitions are given. **Direct prompting** evaluates completion likelihoods normalized over the four answer choices ('people', 'animals', 'birds', 'researchers'). **Chain-of-thought**-like or **auxiliary knowledge** approaches would query a LLM or knowledge base for additional context. Our THINKSUM approach to this task queries a LLM (GPT-2 XL) to produce sets of examples defining the nonce words, then marginalizes over substitutions of these examples into the target phrase.

by prompting GPT-3 with the definition and asking for examples. However, it may also consist of scoring alone, in order to form a matrix of probabilities over which probabilistic inference is performed.

- **SUM** (slow thinking / **SUM**marization / reasoning step): probabilistic inference that aggregates generated strings or probabilities to produce the final answer. The summarization typically involves, and often entirely consists of, summing of probabilities of strings (computed in the THINK step), as in Fig. 1, where the final word is assumed to be sampled from a mixture of possible substitutions of 'binne' and 'bam' words into the input.

THINKSUM is named by analogy with other algorithms with two basic operations that 'expand' and 'aggregate', like MapReduce in distributed computing and sum-product in graphical models.

We discuss different ways to THINK and to SUM in section §2, but we start with one example, illustrated in Fig. 1, motivated by the CONCEPTUAL COMBINATIONS (INVENTED WORDS) task in BIG-bench (Srivastava et al., 2022). In this task, the LLM is provided with two invented words and their definitions in the input. The LLM is then asked to infer the most plausible sentence that uses a combination of the invented words. As the words are invented, they are not common or consistently used in the training set, and the LLM needs to understand and combine the definitions of the invented words to reason about the meaning of the combination. The LLM is queried to produce example instances of the invented words with the help of the definitions. These example instances can be substituted into the query in place of the invented words. In this way, by mapping individual spans of the text of interest to sets we arrive at a mixture model (in this example, a mixture with 25 components, for 5 possible replacements of each of the words), which can be used in the same manner the original LLM is used, either to score text or to generate it token by token. In this case, when we score all candidate completions using this mixture model and normalize over the four choices, the correct answer – that 'binne bams' are for animals and not people – becomes most likely.

An important difference between THINKSUM and existing chain-of-thought-like prompt engineering methods (Wei et al., 2022; Kojima et al., 2022), is that the reasoning step is not reduced to a generation problem for the LLM, but is performed as a probabilistic inference external to the LLM. This reduces its vulnerability to features of the prompt, such as accidental distraction of the LLM by spurious patterns. Instead, we engineer the slow thinking process to make parallel calls to the LLM to query for intermediate information, then possibly perform programmatic recombination of strings (THINK). The final reasoning step – in which likelihoods obtained from the LLM for the recombinations derived from earlier steps of the reasoning process are combined to make the final prediction

– is left to classical probabilistic reasoning (**S**UM). In a sense, **S**UM replaces the self-attention mechanism over linear text, which is used as the sole 'reasoning' mechanism in chain-of-thought-like approaches that expect the intermediate 'thoughts' to take the form of generated tokens intervening between the input and output. Fig. 1 shows the potential brittleness of such 'reasoning', especially in smaller models, which have stronger recency bias (Malkin et al., 2022): if we simply list generated examples as additional context in the prompt, the recency bias causes the LLM to still give higher probability to 'people' than to 'animals', simply because 'bam' (simple dwelling) examples are given after the 'binne' examples.

Imposing an alternative reasoning system over an associative "knee-jerk reaction" system has an analogy with models of human cognitive processes (Tversky & Kahneman, 1974; Kahneman, 2011) that separate System 1 (fast thinking) and System 2 (slow thinking). System 2 acts as a 'controller' that can prime System 1 to appropriately bias its fast thinking, and, in the context of reasoning with deep learning models, has been interpreted as operating with sparse concepts that can be described in language (Bengio, 2017; Goyal & Bengio, 2020). Through repeated usage, the slow-thinking functions of System 2 can become efficiently compressed into System 1 intuitions, in the same manner that iterative 'reasoning' functions of which smaller LLMs are not capable become zero-shot generation capacities for large LLMs.

As is the case with humans, there is always the next frontier of problems where a trained model with remarkable 'intuition' needs to be slowed down. The main claim of this paper is that more is possible with LLMs of existing scale when they are used in concert with a wise System 2 controller that allows for probabilistic inference.

## 2 FAST AND SLOW THINKING WITH LLMS

The example in Fig. 1 falls into a general **T**HINK**S**UM paradigm which extends the reasoning capabilities of a given model by explicitly associating certain *text spans* with *sets of other strings*, which may serve as alternatives or elaborations/explanations and can be either defined by the user or inferred by the LLM itself. These associations then provide multiple texts to be evaluated, again by the LLM itself. The collection of resulting probabilities provides an opportunity to summarize the text using standard probabilistic inference techniques, which usually include a summation.

### 2.1 HOW TO **T**HINK

Here we list examples of the "fast thinking" that precedes the summarization stage.

**Elementary string manipulations.** Standard ways to turn a question into a prompt that can be given to a LLM for generation or scoring involve choices (e.g., of the prompt format) that can be seen as being made by a controlling agent. The standard approach to multiple-choice questions is to write them as Cloze tasks. However, there are nontrivial operations used in inference procedures that sometimes work better, such as:

- **Order inversion**: Exchanging the order of the question and answers, as in Min et al. (2022).
- **Premise erasure**: Deleting a part of the question. Removing a premise with which the answer is expected to have high mutual information is a step in inference procedures that aim to correct for bias towards answers with high unconditional likelihood (Zhao et al., 2021; Holtzman et al., 2021; Malkin et al., 2022).

**Substitution and normalization.** An example is shown in Fig. 1. Elements from a set may be substituted in place of 'slot' words in a prompt, such as 'cat' substituted for 'binne' in the prompt "`A binne bam is a place for`". This operation can be combined with syntax-normalization steps that are reliably achieved by standard NLP tools, such as ensuring subject-verb agreement.

**Example and list generation.** A LLM can be prompted to generate or score lists of words or phrases. We suggest and experiment with three instances of this:

- **Example generation**: In Fig. 1, the LLM is prompted to turn a definition or characterizing property, such as 'simple dwelling', into a list of examples. This can be achieved with a prompt such as "`A bam is a simple dwelling.  Examples:  1.`". The generated completion can be parsed into a set to be used later in the inference procedure.
- **List extension**: A similar approach can also be used to hallucinate additional possible answers to questions, as we will show in some of the experiments.
- **List of words**: Similar prompts provide an even simpler **T**HINK method that we use for scoring – but not generation – in several tasks. Just prompting a LLM with "`List of words:  A,`"

*B*", where *A* and *B* are words or phrases, and computing the likelihood of *B* conditioned on "`List of words:` *A,*" is a good measure of semantic relatedness of *A* and *B*.

**Fact generation.** This way of **THINK**ing associates an input word with a set of phrases, in a similar manner to generating examples from a definition. It can be achieved with prompts such as "`List facts about cats. 1.`" The generated facts are good targets for substitutions of other concepts ('dogs', 'galaxies') in place of the concept ('cats') about which facts are generated. A variation on this asks the LLM to generate differences between two concepts, as shown in Fig. 2 (right).

**Translation.** The LLM can be prompted to convert between different forms of representing the same concept as a sequence of tokens. We use two basic examples of this in experiments:

- Translation between languages by prompting the LLM in formats such as "`French: J'adore les chats noirs. English:`". A very similar approach can be used to convert non-alphabetic symbols, such as emoji, into words with a similar meaning.
- Converting text to formal (symbolic) structures, like turning a word problem into a collection of mathematical equations.

### 2.2 HOW TO **SUM**

**Elementary inference.** As above, we begin by listing existing standard ways of turning LLM outputs into answers, which we see as trivial cases of aggregation (**SUM**).

- **Posterior computation** by normalizing probabilities over a set.
- **Majority/minority vote (argmin/argmax)**: a component of most answer selection procedures.
- **Thresholding**: Used when an answer depends on the value of a probability (or a difference or ratio of probabilities). This can be used when a discrete answer needs to be produced from a real-valued output likelihoods.
- **Ratio of likelihoods**: Likelihoods from different variants of the same prompt can be combined by considering their ratio or more general log-linear or other mixture. For example, this can be done to correct the likelihood of an answer conditioned on a question by its unconditional likelihood, in combination with the **Premise erasure** operation described above.

**Mixture (average) aggregation.** A collection of prompts can be treated as the components of a mixture model over completions. An example is shown in Fig. 1, where substitutions of a set of words yield 25 different prompts. Likelihoods of the completion over these 25 prompts are averaged.

**Product aggregation.** We use products of likelihoods in two different ways:

- In a similar way as mixtures, but when the more natural probabilistic model has *all* elements of a set (of prompts) generating the answer, such as when a description or definition must be satisfied by all concepts in a set.
- In a task where we are to determine whether a statement *S* or its negation *S'* is true, we can compute the likelihood of both *S* and *S'* being true (as posterior over the tokens 'True' and 'False' in an appropriate prompt), then compare $p(\texttt{True}|S)p(\texttt{False}|S')$ (*S* is true and *S'* is false) with $p(\texttt{False}|S)p(\texttt{True}|S')$ (*S* is false and *S'* is true).

**Fitting latent variable models.** See §C for an example of fitting a discrete latent variable (clustering) model over the likelihoods produced by a LLM.

## 3 EXPERIMENTS

We compare **THINKSUM** with the published results for *n*-shot GPT-3 175B (davinci) in BIG-bench, where $n \in \{0, 1, 2, 3\}$. Our main results are shown in Table 1. Below, we describe **THINKSUM** for each task. Detailed descriptions are in §B, and examples of each task appear in Table D.1.

### 3.1 SEMANTIC RELATEDNESS

#### 3.1.1 INTRODUCTORY EXAMPLES: PHRASE RELATEDNESS AND CODENAMES

**Phrase relatedness.** Each question in the multiple-choice PHRASE RELATEDNESS task requires to determine which of a given set of words or phrases $\{w_i\}$ is related to a query phrase *q*. We query the LLM for the likelihood of *q* following a **List of words** prompt to form a vector of likelihoods:

$$p_i = p_{\text{LLM}}(q \mid \text{``List of words: } w_i, \text{''}).$$

The answer selected is the one with highest likelihood, $\arg\max_i p_i$ (a trivial **SUM** operation). We note that this is also an instance of **Order inversion**: the query is scored following a prompt in which each of the candidate answers is substituted.

Table 1: Standard metric (BLEU for CODENAMES, accuracy for other tasks) for GPT-3 175B (davinci) and **THINKSUM** with 175B (davinci), InstructGPT and GPT-2 XL on BIG-bench tasks.

| Task | Avg. H | GPT-3 (davinci) $n$-shot | | | | THINKSUM | | |
| | | $n = 0$ | 1 | 2 | 3 | GPT-3 | InstructGPT | GPT-2 XL |
|---|---|---|---|---|---|---|---|---|
| PHRASE RELATEDNESS (§3.1.1) | 0.74 | 0.37 | 0.42 | 0.52 | 0.59 | 0.85 | **0.87** | 0.79 |
| CODENAMES (§3.1.1) | 0.18 | 0.01 | 0.11 | 0.16 | 0.19 | 0.37 | **0.41** | 0.36 |
| ODD ONE OUT (§3.1.2) | 0.80 | 0.27 | 0.20 | 0.23 | 0.23 | 0.80 | **0.84** | 0.71 |
| NOVEL CONCEPTS (§3.2.1) | 0.67 | 0.47 | 0.47 | 0.56 | 0.56 | 0.72 | **0.75** | 0.50 |
| INVENTED WORDS (§3.2.2) | N/A | 0.29 | 0.14 | 0.14 | 0.21 | 0.64 | **0.71** | 0.29 |
| SPORTS UNDERSTANDING (§3.3.1) | 0.71 | 0.50 | 0.50 | 0.50 | 0.50 | 0.71 | **0.74** | 0.54 |
| KNOWN UNKNOWNS (§3.3.2) | **0.80** | 0.61 | 0.52 | 0.48 | 0.50 | 0.54 | 0.76 | – |
| MISCONCEPTIONS RUSSIAN (§3.4.1) | 0.65 | 0.33 | 0.33 | 0.41 | 0.35 | **0.70** | 0.61 | – |
| EMOJI MOVIE (§3.4.1) | **0.93** | 0.12 | 0.18 | 0.12 | 0.19 | 0.80 | 0.75 | – |
| FIVE OBJECTS (§3.4.2) | N/A | 0.23 | 0.29 | 0.28 | 0.32 | – | **0.77** | – |

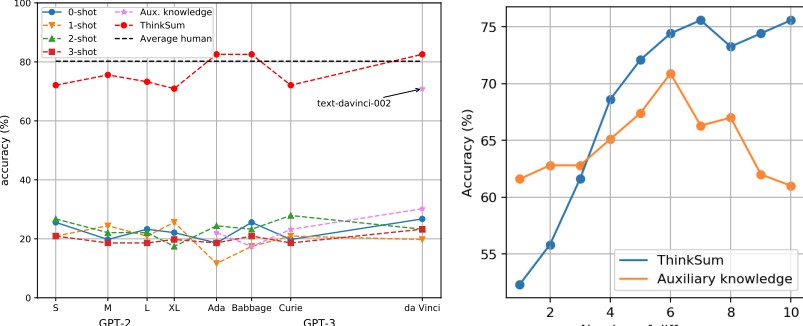

Figure 2: ODD ONE OUT. **Left:** Performance of GPT-3 ($n$-shot, $n = 0, 1, 2, 3$), auxiliary knowledge, and **THINKSUM** with various model sizes. **Middle:** Auxiliary knowledge vs. **THINKSUM** with varying number of differences. **Right:** Prompt used to generate knowledge statements.

**Codenames.** Each question in CODENAMES requires selecting the $k$ words from a set $\{w_i\}$ that are most closely related to a query word $q$. We form a vector $p_i$ in the same way as for PHRASE RELATEDNESS, then select the top $k$ entries in $p_i$ to produce the output.[1]

### 3.1.2 ODD ONE OUT

We examine possible **THINK** and **SUM** approaches in greater depth on the ODD ONE OUT task, in which the word in a set $W = \{w_i\}$ that is *least* semantically related to the others must be chosen.

**List of words.** We first consider an approach similar to that used in §3.1.1. We form a matrix $P_{ij}$ using a **List of words THINK** prompt for each pair of indices $i, j$:

$$P_{ij} = p_{\text{LLM}}(w_j \mid \text{``List of words: } w_i, \text{ ''}).$$

This matrix is aggregated by averaging over $j$ (in log domain) and selecting the $i$ with lowest average, i.e., least likelihood of being generated by a product mixture of all words in the set: $i = \arg\min_i \prod_j P_{ij}$. This is a case of the **Product aggregation** operation.

Because this approach is the most successful with all model sizes we experimented with, its performance is reported in Table 1. Remarkably, near-average-human accuracy is maintained for all model sizes from GPT-2 Small to the largest GPT-3 model (Fig. 2 (left)).

**Fact generation.** As an alternative approach, we use a **Fact generation** prompt. An effective way to mine facts for semantic relatedness tasks is to consider two items in the same context in order to get relevant facts regarding how items are related to each other (prompt in Fig. 2 (right)). The demonstration used in the prompt ensures that the LLM generates statements in an expected format, which can be parsed and used for probability computation later. Using this prompt, we obtain a collection of statements $S = \{s_i\}$ about items $w_j$. We treat each generated $s_i$ as a template into

---

[1]Because the task is evaluated by BLEU score against the reference answers listed in alphabetical order, we perform the additional step of converting the top indices to the answer in the right format. Alphabetization of short lists is trivial in code, but can also very reliably be done by prompting GPT-3.

which different words $w$ can be substituted and denote by $s_i\langle w \rangle$ the **Substitution** of word $w$ into template $s_i$. We then form a $|S| \times |W|$ matrix $P_{ij}$, defined by $P_{ij} = p_{\text{LLM}}(s_i\langle w_j \rangle)$. Then, we can perform **Minority voting**: we take argmin over $j$ and pick as the answer the most frequently occurring value, i.e., the item that is most often the least likely to fit a generated statement.

**Comparison with auxiliary knowledge approaches.** We compare our method with a knowledge-based prompting method, herein referred to as auxiliary knowledge. In auxiliary knowledge, we prepend generated facts in the prompt before the question. Details of the prompt for auxiliary knowledge are provided in §D.2. In Figure 2 (middle), we show that the accuracy of **Fact generation**-based THINKSUM rises as the number of generated facts is increased, while the auxiliary knowledge technique peaks and then degrades as the prompt lengthens.

Fig. 2 (left) shows how performance varies with the size of the LLM used for GPT-3, auxiliary knowledge and THINKSUM on ODD ONE OUT. Even with GPT-2 Small, THINKSUM dramatically improves over much larger largest zero- or few-shot models with or without auxiliary knowledge. The latest iteration of the largest GPT-3 model, text-davinci-002, is the only model variant that, with the help of auxiliary knowledge, achieves competitive performance with THINKSUM. This result provides experimental evidence for our claim that while new models may create qualitative jumps, THINKSUM can push the performance limits of smaller model variants through slow thinking. Additional experiments on auxiliary knowledge are provided in §C.

## 3.2 SUBSTITUTION AND AGGREGATION

### 3.2.1 NOVEL CONCEPTS

In the multiple-choice NOVEL CONCEPTS task, a set of words or phrases $W = \{w_i\}$ and a set of statements $S = \{s_j\}$ with third-person plural pronoun subjects ('They all...') are given, and the statement which is true for all items in $W$ must be determined.

We treat each statement $s_j$ as a *template*, into which words $w$ can be substituted by replacing 'They all' with $w$. Denoting by $s_j\langle w \rangle$ the substitution of $w$ into $s_j$, we form a $|W| \times |S|$ matrix $P_{ij}$ by scoring the **Substitution** of each word into each statement and considering the **Ratio of likelihoods** with the template without substitution: $P_{ij} = \frac{p_{\text{LLM}}(s_j\langle w_i \rangle)}{p_{\text{LLM}}(s_j)}$. We then perform **Product aggregation** to select the statement which is most likely to be generated by all words in the set. To be precise, the selected statement is $\arg\max_j \prod_i P_{ij}$.

### 3.2.2 INVENTED WORDS

In INVENTED WORDS, two nonce words $x_1, x_2$ are defined and the correct statement must be chosen out of a set of statements $S = \{s_j\}$ that begin with (possibly inflected forms of) "$x_1\ x_2$" (Fig. 1).

We use an **Example generation** prompt to obtain a set of example words fitting the definitions of $x_1$ and $x_2$. We thus obtain sets $S_1$ and $S_2$ of words that can be substituted for $x_1$ and $x_2$, respectively.

We treat each statement $s_j$ as a template into which words $w_1 \in S_1$ and $w_2 \in S_2$ can be substituted by replacing $x_i$ with $w_i$ and normalizing the syntax to ensure subject-verb agreement. Denoting by $s_j\langle w_1, w_2 \rangle$ such a substitution, we form a vector of probabilities $p_j$ by scoring the **Substitution** of each possible pair of words into each statement and performing **Mixture aggregation** and considering the **Ratio of likelihoods** with the template without substitution:

$$p_j = \frac{\frac{1}{|S_1||S_2|} \sum_{w_1 \in S_1, w_2 \in S_2} p_{\text{LLM}}(s_j\langle w_1, w_2 \rangle)}{p_{\text{LLM}}(s_j)}.$$

The statement $s_j$ with highest likelihood under this normalized mixture, $\arg\max_j p_j$, is selected.

## 3.3 UNCERTAINTY AND HALLUCINATION DETECTION

LLMs are prone to generating hallucinations that contain incorrect statements. The likelihoods of these statements are often dominated by short plausible patterns, which also makes it difficult for LLMs to evaluate their own uncertainty about a fact. Thus, detection (Liu et al., 2021; Zhou et al., 2021) and reduction of such hallucinations is crucial for widespread use of LLMs in real applications. (Dziri et al., 2021; Shuster et al., 2021).

### 3.3.1 SPORTS UNDERSTANDING

Questions in SPORTS UNDERSTANDING ask to determine whether it is 'plausible' or 'implausible' that a professional sports player $x$ (e.g., 'Draymond Green', a basketball player) performed an action

*a* associated with a sport (e.g., 'threw a touchdown', an action in American football). It is implied that the combination of *x* and *a* is plausible if the sport with which player *x* is associated coincides with the sport in which action *a* is performed. We consider an approach that does not rely on identifying the latent variable (sport) as an intermediate step and is thus more generalizable to other domains.

We use an **Example generation THINK** prompt to produce a set *S* of players who perform action *a*, then do **Posterior computation** by normalizing the likelihood assigned by the LLM to each player in *S*, as well as *x*, performing action *a*:

$$\forall y \in S \cup \{x\} \quad p(y|a) = \frac{p_{\text{LLM}}(\text{``}y\ a\text{''})}{\sum_{y' \in S \cup \{x\}} p_{\text{LLM}}(\text{``}y'\ a\text{''})}$$

The statement is considered to be implausible if the posterior on *x* is sufficiently low (**Thresholding**) – see Fig. 3.

### 3.3.2   KNOWN UNKNOWNS

Questions in the KNOWN UNKNOWNS task ask to determine whether the answer to a question is a certain precise concept or 'unknown'.

Given a question *q* (e.g., "What was the temperature in Cuzco on the day of the Emperor Vespasian's birth") and the candidate precise answer *a* (e.g., 25°C), we use a **List extension** prompt to generate a set *S* of other possible answers to *q*. We then do a **Posterior computation** over *S* and the original answer *a*, similar to that used for SPORTS UNDERSTANDING:

$$\forall y \in S \cup \{a\} \quad p(y|q) = \frac{p_{\text{LLM}}(\text{``}q?\quad y\text{''})}{\sum_{y' \in S \cup \{a\}} p_{\text{LLM}}(\text{``}q?\quad y'\text{''})}.$$

The answer *a* is chosen if the posterior on *a* is sufficiently high (**Thresholding**), and otherwise 'unknown' is chosen.

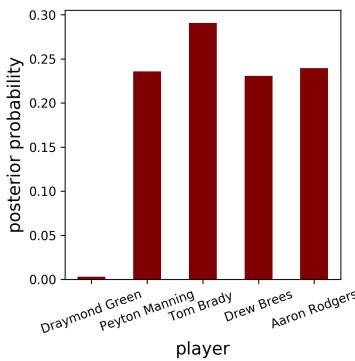

Figure 3: Example posterior probabilities generated from text-davinci-002 for SPORTS UNDERSTANDING with the description *"threw a touchdown"*. The basketball player given in the question *Draymond Green* has a much lower posterior probability than the generated football players, from which we conclude the sentence *"Draymond Green threw a touchdown."* is implausible.

### 3.4   TRANSLATION

#### 3.4.1   TRANSLATING BETWEEN LANGUAGES AND WRITING SYSTEMS

**Russian misconceptions.**   In the MISCONCEPTIONS RUSSIAN task, the true statement must be chosen out of a pair of Russian sentences: a statement *s* and its negation *t*.

We first describe an approach that does not use translation and already performs better than random guessing – and better than baseline methods that simply select the more likely of the two statements – using the largest GPT-3 model, which has sufficient knowledge of Russian. We compute the posterior over the two hypotheses "*s* is true, *t* is false" and "*s* is false, *t* is true":

$$p_{\text{LLM}}(\text{``T''} \,|\, \text{``T or F? } s. \quad \text{Answer: ''}) p_{\text{LLM}}(\text{``F''} \,|\, \text{``T or F? } t. \quad \text{Answer: ''}),$$

$$p_{\text{LLM}}(\text{``F''} \,|\, \text{``T or F? } s. \quad \text{Answer: ''}) p_{\text{LLM}}(\text{``T''} \,|\, \text{``T or F? } t. \quad \text{Answer: ''}).$$

where T denotes True and F False in the actual prompt. This is a kind of **Product aggregation**. If the posterior on the first option is higher, *s* is chosen as the true statement; otherwise, *t* is chosen.

This approach can be combined with a **Translation** prompt that produces translations of *s* and *t* into English, then uses these translations in place of *s* and *t* in the above computations. The approach can be further extended by sampling a *set* of translations and performing **Mixture aggregation** over the translations. Our reported result uses 10 generated translation for each statement, but it is only 2% higher than the result using one generated translation.

**Emoji movie.**   The multiple-choice EMOJI MOVIE task requires selecting the name of a movie from a list $\{m_i\}$ that is best described by a sequence of emoji symbols $s = (s_1 \ldots s_n)$. An **Order inversion** prompt performs best on this task using the Davinci variant of GPT-3: choosing the answer

$$\arg\max_i p_{\text{LLM}}(s \,|\, \text{``Emoji describing the movie } m_i\text{''}).$$

We also attempt to use a **Translation** prompt to obtain a single-word English description $w_j$ of each emoji $s_j$ in *s*, then score using

$$\arg\max_i p_{\text{LLM}}(w_1 \ldots w_n \,|\, \text{``Words describing the movie } m_i\text{''}).$$

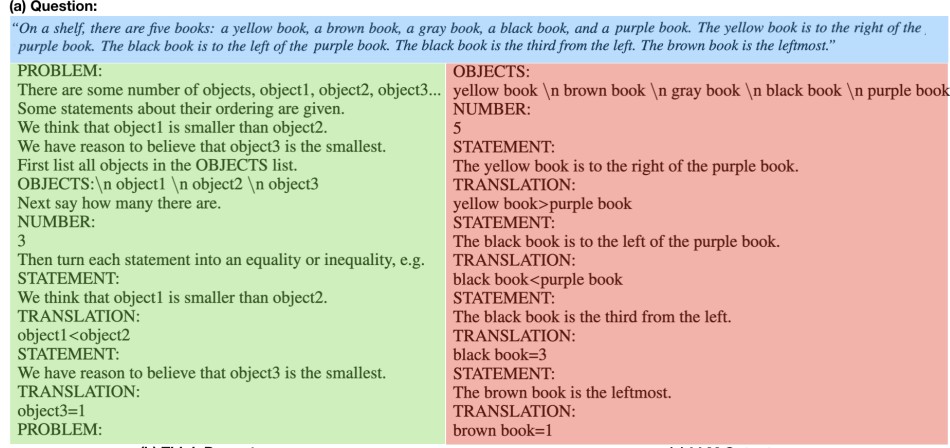

Figure 4: Details for LOGICAL DEDUCTION. (a) Example question from the task, (b) demonstration for the **THINK** prompt, (c) example LLM output.

This approach performs slightly better than **Order inversion** alone using InstructGPT. However, it does not work with the base GPT-3 models, which do not as reliably translate emoji to English.

### 3.4.2 LOGICAL DEDUCTION

In the LOGICAL DEDUCTION task, different types of items and clues regarding their placement are provided, as shown in Fig. 4(a). The goal is to select the correct statement from a set of statements about their placements. The task creators emphasize that this requires parsing information about multiple objects and their relationships, understanding rules regarding ordered objects in various scenarios, and iteratively applying of these rules. The LLM calls in the **THINK** stage of **THINKSUM** can perform mappings required to parse information and understand rules, and the **SUM** stage can integrate mappings of objects to the placements under these rules. Here, we can use a **Translation** prompt to map the given problem into a set of mathematical (in)equalities (Fig. 4(c)).

A **Translation** prompt as in Fig. 4(b) containing generic ordering statements and object names that are not used in the task is sufficient to perform the translation from natural language to mathematical equations, as shown in Fig. 4. To solve the problem, we replace the problem statement with its translation $T$ consisting of (in)equalities, and map each of the objects to the set of strings corresponding to numbers from 1 to $N$, where $N$ is the number of objects. This is accomplished by appending a given problem to the fixed prompt in Fig. 4(b), which acts as a demonstration for one-shot in-context learning. The ordering problems involve a variety of types of objects (cars, birds, etc.) and types of orderings (by size, price, contest ranking, etc.). For a particular problem in Fig. 4(a), we show the returned text from InstructGPT in Fig. 4(c), showing that the demonstration used in the **THINK** prompt generalizes from objects ordered by size to books ordered by position.

Once a translation of the problem into a set of inequalities is obtained, the **SUM** stage considers all possible mappings of items to indices to determine the mapping compatible with the discovered set of (in)equalities. This can be done by an external algorithm or with the LLM itself, as an LLM may be capable of understanding that, for example, "2>3" is a less likely string than "2>1" (see Fig. B.1).

The probability of a target statement like *"yellow book=2"* can thus be obtained by:

$$p(\text{``yellow book=2''} \mid T) \propto \sum_{\mathbf{b} \in \{1,\dots,N\}^N} p_{\text{LLM}}(\{T_t \langle \mathbf{b} \rangle : T_t \in T\} \cup \{\text{``yellow book=2''} \langle \mathbf{b} \rangle\}) \quad (1)$$

where $\mathbf{b}$ denotes the vector of placements for the $N$ items, $T = \{T_t\}_{t=1}^N$ is obtained from the **Translation** prompt as a set of strings, and $s\langle \mathbf{b} \rangle$ denotes the substitution of the corresponding entry in $\mathbf{b}$ in place of the object name in the string $s$. The term inside the sum is a case of **Product aggregation**: the LLM likelihoods of all strings in the set are multiplied together.

In summary, our solution to this task involves composition of two **THINK** operations – a **Translation** into a set of equations and then **Substitution** of numbers in place of item names – and two

SUM operations – a **Product aggregation** followed by a **Mixture aggregation**. (Other options are discussed in §C.)

**Results and discussion.** For the 500 LOGICAL DEDUCTION problems with $N = 5$ objects, THINKSUM yields an accuracy rate of 77% (see Table 1), besting the average human performance. When the necessary summations become large, it becomes very unlikely that pure prompt engineering can be competitive, as even humans need paper and pencil to create and attend to many alternative solutions, and would likely translate the premises into a simpler notation using a single letter (representing a variable to which a numeric value can be assigned) to represent each object, rather than directly attending to facts in the problem statement.

We also tested an auxiliary knowledge method akin to chain-of-thought reasoning, where the information obtained with the prompt in Fig. 4 is appended to the LLM input. In particular, the problem, together with its translation into inequalities, is used as a prompt to each of the answer options, and then the option with the highest likelihood is chosen for the answer. This approach does improve over straightforward zero-shot GPT-3 scoring, but only raises the accuracy to 50% (see Table B.1).

## 4  RELATED WORK

**Improvements to LLM inference.** After the discovery of the in-context learning abilities of LLMs, there has been an explosion of interest in improving inference with LLMs in the zero-shot and few-shot setting Brown et al. (2020); Chowdhery et al. (2022); Rae et al. (2021). One approach to improving the reasoning abilities of LLMs involves appending, or learning to generate, auxiliary knowledge within the prompt (Shwartz et al., 2020; Zelikman et al., 2022; Nye et al., 2021a). Recently, more general auxiliary knowledge or chain-of-thought prompting methods have been proposed (Wei et al., 2022; Wang et al., 2022b; Zhou et al., 2022; Creswell et al., 2022; Wang et al., 2022a; Liu et al., 2022b). Later, Kojima et al. (2022) showed zero-shot chain-of-thought prompting can improve performance on a variety of reasoning tasks. This method does not require any hand-crafted few-shot examples, which is a shared property with THINKSUM. (Nye et al., 2021b) observed that a dual-system approach where an associative "System 1" and a logical "System 2" can increase coherence of LLMs in tasks such as robust story generation and grounded instruction following. The two-step paradigm in THINKSUM is similar, where "System 1" is the (querying of the LLM for) fast thinking, and "System 2" is the probabilistic inference step.

**Brittleness of chain-of-thought prompting.** Despite the recent success of chain-of-thought approaches, recent studies have raised concerns regarding the limitations of chain-of-thought approaches. Webson & Pavlick (2022) observed that instructive prompts perform similarly with misleading or intentionally irrelevant prompts. Additionally, Ye & Durrett (2022) showed improvements due to few-shot chain-of-thought are not observed in question answering, or natural language inference. More critically, few-shot prompts are highly sensitive to the order in which the samples are provided, the prompt format, and the selection of in-context examples, (Lu et al., 2022; Zhao et al., 2021). Thus, it is crucial to design techniques that are robust to such changes in the prompt.

**Probabilistic inference.** More recent approaches have proposed probabilistic inference approaches for tackling true/false question answering and commonsense question answering (Jung et al., 2022; Liu et al., 2022a). Xie et al. (2021) presents a Bayesian inference perspective on in-context learning, and Dohan et al. (2022) formalizes and unifies existing prompting techniques in a probabilistic framework. Our work generalizes such approaches to perform arbitrary probabilistic inference outside of the LLM.

## 5  CONCLUSION

In this paper we presented THINKSUM, a two-step probabilistic inference paradigm that reasons over sets in a structured manner. The fast thinking stage of THINKSUM allows elementary string manipulations as well as natural language prompting, which may enable numerous approaches to solve a natural language task. Even with far smaller model variants, THINKSUM achieves state-of-the-art results on ten difficult tasks in BIG-bench using GPT-family models. The two-step paradigm allows operating over sets instead of manipulating the prompt itself, preventing sensitivity to prompt format during the probabilistic inference in THINKSUM, which is performed outside of calls to the LLM. As a result, THINKSUM is more robust to prompt design, yields more interpretable predictions, and can be combined with many probabilistic inference approaches to tackle a diverse set of tasks.

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

## A  BIG-BENCH LITE

Figure A.1 shows the performance margin between an average human and zero-shot GPT-3 on tasks in BIG-bench Lite, a select subset of tasks chosen by the authors of the benchmark to showcase the most important aspects of LLMs that need improvement. The vertical black bar separates the dataset into tasks where GPT-3 is already within the margin of just 10% compared to the average human accuracy, and the harder tasks (on the left). We show in the main text that some of these harder tasks, in particular EMOJI MOVIE, CONCEPTUAL COMBINATIONS, KNOWN UNKNOWNS, NOVEL CONCEPTS, MISCONCEPTIONS RUSSIAN and LOGICAL DEDUCTION, the margins are shrunk considerably, often exceeding average human performance. Other tasks in BIG-bench lite such as LOGIC GRID PUZZLE and SYMBOL INTERPRETATION share a similar structure to the addressed by **THINKSUM**, and thus could be investigated as part of future work. Another example where **THINKSUM** can be applied is the CODE LINE DESCRIPTION task, where we observe in our preliminary experiments that a simple order inversion can significantly outperform average human accuracy.

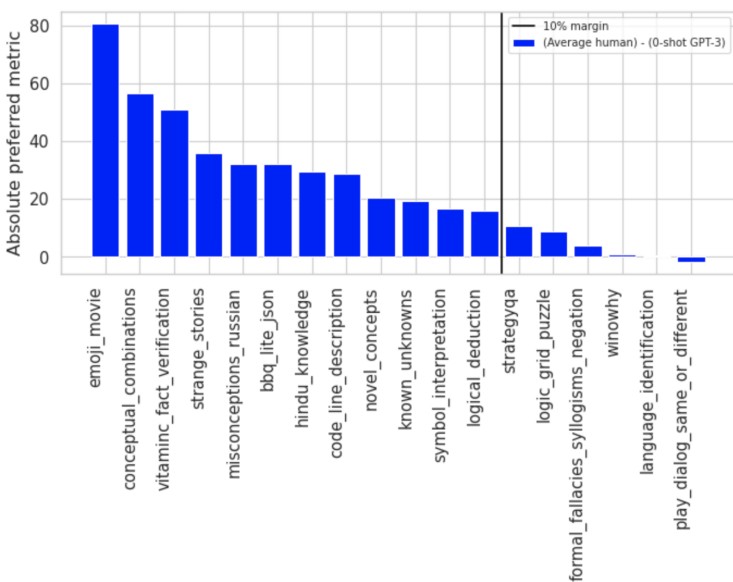

Figure A.1: Margin between 0-shot GPT-3 and average human performance for BIG-bench Lite tasks. Using **THINKSUM**, we address many of the tasks that have greater than 10% performance margin with average human, and significantly reduce and often overturn the margin.

## B ADDITIONAL TASK DESCRIPTIONS

### B.1 SEMANTIC RELATEDNESS

**Phrase relatedness.** In this task, an input phrase, and four target phrases are presented to the language model. The language model is asked to identify the most related choice from the listed target options. An example from the task is *"For each word or phrase, identify the most related choice from the listed options. \n Input: home town \n Option: town center \n Option: location \n Option: native city \n Option: home run"*

**Codenames.** In this task, the language model is asked to identify words associated with a given word. An example from the task is *"Try to identify the 2 words best associated with the word WHITE from the following list: \n book, anchor, rainbow, shoulder, tunnel, sack, drum, pacific, page, mark, gear, glacier. Give your answer in alphabetical order."*

**Odd one out.** This task is aimed at evaluating the capability of LLMs in semantic relatedness. This task presents the model with four to six words, where all words except one word are semantically or grammatically related to each other. The goal for the language model is to identify the odd word. An example question from the task is *"Pick the odd word out: glass, head, arm, leg, hand, foot"*.

### B.2 HALLUCINATION DETECTION

**Known unknowns.** In this task, the aim is to measure the ability of LLMs to identify whether the answer to a question is known, or unknown. If the answer to the question cannot be known, an imaginary answer is generated, and the model is asked to choose between the imaginary answer and the word "Unknown". If the question has a correct answer, the correct answer and "Unknown" are provided as options. An example question from this task is *"How often did Abraham Lincoln cut his toenails?"*, and the options are *"Every Saturday night","Unknown"*. Similar to the sports understanding task, we can use an LLM to generate other plausible answers with the following prompt: *How often did Abraham Lincoln cut his toenails? A possible answer is: \n 1. Every Saturday night. List 4 other possible answers in the same format as the first: \n 2.* Then, the answer given in the question is predicted to be "known" if its posterior is higher by the second most likely option by some margin. In our experiments, we chose this value to be $1/N_e$ where $N_e$ is the number of examples, including the original option.

**Sports understanding.** In this task, artificial sentences are constructed using the name of professional sports players and actions from particular sports. The model is then asked to identify whether

Table B.1: **THINKSUM** vs. auxiliary knowledge.

|  | ODD ONE OUT | PHRASE RELATEDNESS | LOGICAL DEDUCTION ($N = 5$) |
|---|---|---|---|
| **THINKSUM** | 0.84 | 0.87 | 0.77 |
| Auxiliary knowledge | 0.71 | 0.75 | 0.50 |

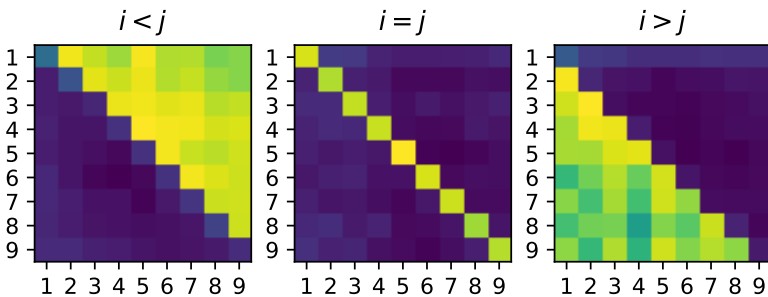

Figure B.1: Probabilities of different (in)equalities according to GPT-3 text-davinci-002 (logit).

the sentence is plausible, where a sentence is considered plausible if the sport of the player matches the sport of the action described in the sentence. An example from the task is *"Statement: Draymond Green threw a touchdown. Plausible/implausible?"*

For **THINKSUM** to be able to parse outputs, GPT-3 generations need to be in a pre-determined format. While larger models can obey a format without demonstrations, for smaller models it is helpful to demonstrate the format with an example. Thus, we use the following prompt: *"List 4 examples of players who scored a rabona goal.\n 1. Cristiano Ronaldo\n 2. Erik Lamela\n 3. Mario Balotelli\n 4. Angel Di Maria\n List 4 examples of players who threw a touchdown.\n1."*.

### B.3 CONCEPT UNDERSTANDING

In the following tasks, the shared goal is to test the ability of LLMs on concepts over entities that have likely not been observed during training.

**Conceptual combinations: Invented words.** In this task, the LLM is provided with two invented words, and their definitions in the input. The LLM is then asked to infer the most plausible meaning resulting from the combination of the invented words. As the words are invented, they are not present in the training set, and the LLM needs to understand and combine the definitions of the invented words to reason about the meaning of the combination. An example is: *"The word 'binne' means any animal that is furry and has four legs, and the word 'bam' means a simple sort of dwelling. Question: Which of the following sentences best characterizes binne bams?"*. Similar to SPORTS UNDERSTANDING, we can use the following prompt to force the LLM to obey a fixed format: *"List synonyms of binne, separate synonyms by comma:"*

**Novel concepts.** In this task, the LLM is presented with two to four disparate entities that typically would not co-occur frequently, but share an underlying conceptual or linguistic concept. The aim is to test the ability of the LLM to reason about entities that are unlikely to have been observed in the same context during training. In a multiple-choice setting, the LLM is given concepts relating to the entities, and is asked to generate the intended concepts against carefully chosen tempting distractors. The choices are not presented in the prompt. An example question from the task is as follows: *"What do the following have in common? 1) bumble bees 2) 01010101 3) race cars"*, and the answer options are *They all make noise, "They all are yellow, They all are binary, They all go fast, They all have stripes"*.

### B.4 TRANSLATION

**Emoji Movie.** In this task, the LLM is presented with emojis that describe the plot of a movie, and the model is asked to guess the correct movie from the given options. The task evaluates the LLMs understanding of emojis, and its knowledge of popular movies. An example from the task is

Table C.1: Different alternatives of probabilistic reasoning with **THINKSUM** for solving ODD ONE OUT.

| Model | List of words | Latent variable model | Minority voting |
|---|---|---|---|
| **THINKSUM** davinci-002 | 0.84 | 0.67 | 0.70 |
| **THINKSUM** davinci-001 | 0.74 | 0.77 | 0.70 |

*"What movie does this emoji describe? {lying-face-emoji} \n, pinocchio \n heat \n the big sleep, \n inception \n lilo & stitch*

**Misconceptions Russian.** In this task, the LLM is presented with sentences in Russian that contain misconceptions, and their true counterparts. The task evaluates if the LLM believes common misconceptions. One example of a target pair from the task, translated to English is as follows: *"Vaccines do not cause autism. \n Vaccines cause autism."*

## C ADDITIONAL EXPERIMENTS

**THINKSUM vs. auxiliary knowledge.** Table B.1 shows the comparison between auxiliary knowledge and **THINKSUM**. All methods used text-davinci-002. For PHRASE RELATEDNESS, auxiliary knowledge was generated using the "list differences" prompt shown in Figure 2 (right). For both auxiliary knowledge and **THINKSUM**, 6 number of differences were used, as that was the best for auxiliary knowledge, shown in Figure 2 (middle). **THINKSUM** ODD ONE OUT and PHRASE RELATEDNESS are solved with the list of words prompt. For LOGICAL DEDUCTION, the **THINK** prompt shown in Fig4 was included before the question in the prompt. In all cases, **THINKSUM** outperforms auxiliary knowledge.

**Latent variable models for ODD ONE OUT.** While the detection of the odd item can be performed with simple operations, it is also possible to assume that there is a latent structure consisting of two or more clusters such that the facts and items belonging to a cluster can be freely interchanged keeping probability of such combination high. While there are simpler alternative solutions to this task, the latent variable model enables selecting the facts that characterize the majority class, explaining why the minority item is ruled as the odd one out. Thus, expanding on the problem and applying a latent variable model can help interpret the decisions of the system.

More formally, since $I$ includes items that are semantically related, and the odd, we can model $i \in I$ and $f \in F$ to be generated from a latent class $c \in \{0, 1\}$. Then, the conditional distribution can be modeled as:

$$P(i, f|c) = P(i|c)P(f|c) \quad P(i, f) = \sum_c P(i, f|c)$$

The semantic components, groupings $P(i|c)$ and $P(f|c)$ can be computed from the matrix using expectation-maximization (EM; Dempster et al., 1977). Then, the score for an item $i$ belonging to a cluster and all other items $m \in S, \{m \neq i\}$ belonging to another cluster can be found as $S_i = \sum_{c,c' \neq c} P(i|c)P(c) \prod_{m \neq i} P(m|c')P(c')$.

We show the effectiveness of the latent variable models in Table C.1, where we analyze different methods for solving ODD ONE OUT using the InstructGPT variants text-davinci-001 and text-davinci-002. For the "latent variable model" and "minority voting" methods, we use number of differences $N_d = 5$. The latent variable model is run for 200 iterations. All probabilistic reasoning methods perform well, outperforming previous baselines reported in Table 1. Each of these approaches can be applicable in other tasks of similar structure to ODD ONE OUT.

**Using GPT-3 or external algorithms to evaluate inequalities in LOGICAL DEDUCTION.** Fig. B.1 shows the matrix of posterior probabilities evaluated using InstructGPT (text-davinci-002) for strings of form "x=y", "x<y", "x>y" for $x, y \in \{1, .., 9\}$. The probabilities are computed using prompts of the form *"True or false: x<y? The answer is:"* where $x$ and $y$ are substituted with numbers from $\{1, .., 9\}$, and normalizing the probability of the first token over the two options *"true"* and *"false"*. These are the probabilities that statements $T_t$ would evaluate in (1).

Given the capability of InstructGPT to both translate given logical deduction problems into (in)equalities (Fig. 4) and to evaluate each of them after replacement of items with placement

numbers (Fig. B.1), we conclude that it is perfectly within its capabilities to parse and understand the problem, and the **SUM** stage is there simply to go over all possible mappings, the way a human might. But, just as a human might use shortcuts in the search, the **SUM** stage of **THINKSUM** could be implemented in more or less efficient ways. For example, instead of summing over all possible assignments of the five items, we can avoid the ones that are not permutations of $\{1, 2, 3, 4, 5\}$. Furthermore, instead of using $p_{\text{LLM}}$ from Fig. B.1, we can simply evaluate each inequality externally, giving a high constant probability for each statement in $T$ and the target statement where a configuration of item placements makes the statement correct and low constant probability whenever the statement is incorrect. Whichever evaluation mechanism we use, the summing can be aborted whenever an incorrect statement is detected (typically there are four (in)equalities that describe the problem, and one equality statement that describe the option to be evaluated; we want to find a configuration of item placements such that all of these are correct, or have high probability under $p_{\text{LLM}}$.

The prompt in Fig. 4(b) instructs the LLM to assign positive or negative ordinal numbers depending on the language used (the smallest object gets placement 1, while the second largest one gets −2, meaning 'second from the end'). A negative order number $r$ is then turned in our code into $N + r + 1$ before evaluating statements. Sometimes, LLM does not follow this instruction, but simply labels the largest, right-most, most expensive, etc. as $N$, instead of −1, which is equivalent. One possible failure mode of this kind of **THINKSUM** is that the LLM may translate inequality statements inconsistently with equality statements (e.g., by treating the leftmost item as "1", and being consistent with this choice for other equality constraints, but translating inequality constraints consistent with the reverse order, with 'left of' meaning >). This can be dealt with by adding an option to replace placement numbers $r$ in equality statements by $N − r + 1$. This doubles the number of evaluations to be done (as each $T$ now has two versions), but allows for an auto-correction in **SUM**.

### C.1 COMPARISONS WITH CHAIN-OF-THOUGHT APPROACHES

Following Wei et al. (2022), we use "chain-of-thought" to mean LLM scoring approaches that use insertion of generated tokens between the prompt and the target answer. The model is taught, using few-shot demonstrations, how to generate these intermediate tokens. Above we have compared **THINKSUM** with approaches that add extracted (from an auxiliary LM call), not generated (within the LM's linear workspace) token sequences after the prompt, for the ODD ONE OUT, PHRASE RELATEDNESS, and LOGICAL DEDUCTION tasks (see Table B.1).

With suitable examples, it may be possible for a chain-of-thought approach to replace the **THINK** phase, by learning from demonstrations to generate the appropriate knowledge, and parts of the **SUM** phase, although inference over parallel evaluations of the LLM is no longer possible. Our auxiliary knowledge baselines make precisely that generous assumption and focus the comparisons on the need for parallel calls and reasoning over possibilities using probabilistic inference (instead of leaving it to the LLM to make the right conclusions from the list of extracted alternatives).

Although we expect that appending facts in a standard format to the prompt would help the model more than teaching the model to generate these facts, we experimented with chain-of-thought approaches on several tasks. Table C.3 shows example demonstrations and prompt formats used for each task, and Table C.2 shows the results using two variants of the largest GPT-3 model.

Table C.2: Comparison of **THINKSUM** with chain-of-thought prompting approaches.

| Task | GPT-3 (davinci) | | | GPT-3 (davinci-002) | |
|---|---|---|---|---|---|
| | Direct | CoT | **THINKSUM** | CoT | **THINKSUM** |
| ODD ONE OUT | 0.27 | 0.33 | 0.80 | 0.64 | 0.84 |
| PHRASE RELATEDNESS | 0.59 | 0.55 | 0.85 | 0.79 | 0.87 |
| LOGICAL DEDUCTION | 0.32 | 0.25 | – | 0.39 | 0.77 |
| KNOWN UNKNOWNS | 0.61 | 0.70 | 0.54 | 0.74 | 0.76 |
| INVENTED WORDS | 0.29 | 0.50 | 0.64 | 0.64 | 0.71 |

As expected, **THINKSUM** outperforms chain-of-thought prompting on all tasks with all variants except KNOWN UNKNOWNS with the davinci variant, where direct prompting already performs well. (We did not evaluate **THINKSUM** with davinci on LOGICAL DEDUCTION because prompts like the one in Figure 4 did not reliably produce outputs in the correct format; notice that chain-of-thought is barely better than random guessing (20%).)

Figure D.1: Auxiliary knowledge prompting applied to ODD ONE OUT. Facts are generated using the "list differences" prompt described in Figure 2 (right) and post-processed according to §D.2.

When interpreting these results, it is important to note that only one prompt format was evaluated for both chain of thought and **THINKSUM**, and the format of prompts and demonstrations can have a strong and often unpredictable effect on the LLM. We observed that chain-of-thought approaches are highly sensitive to minor changes in the prompt format or the construction of in-context examples, consistent with the known biases of in-context learning when a potentially lengthy prompt is evaluated (Lu et al., 2022; Zhao et al., 2021). On the other hand, using structured, shorter components is more reliable, as demonstrated by the efficacy of the **THINK** prompts used in **THINKSUM**.

# D  ADDITIONAL EXPERIMENTAL DETAILS

Our experiments are performed using four different sizes of GPT-2 (Small, Medium, Large, and XL) (Radford et al., 2019), GPT-3 with four different model sizes (ada,babbage,curie,davinci) (Brown et al., 2020), and InstructGPT (Ouyang et al., 2022). All GPT-3 experiments are run between August 2022 and September 2022 by using the OpenAI API. Our GPT-2 experiments were run in PyTorch (Paszke et al., 2019) and the Hugging Face Transformers library with a Tesla K80 GPU.

## D.1  HYPERPARAMETERS

**Maximum generation length.** For tasks that require **example and list generation**, such as CONCEPTUAL COMBINATIONS, KNOWN UNKNOWNS, and SPORTS UNDERSTANDING, we use max_tokens = 100. For **fact generation** in ODD ONE OUT with auxiliary knowledge and **THINKSUM**, we use max_tokens = 1000.

**Temperature.** All GPT-2 experiments used temperature = 0.5. For SPORTS UNDERSTANDING and translation tasks, we used temperature = 0.5 to promote diversity of generated plausible options. All other experiments used temperature = 0.

**Number of examples** ($N_e$). For CONCEPTUAL COMBINATIONS we used $N_e = 2$, and for KNOWN UNKNOWNS and SPORTS UNDERSTANDING we used $N_e = 4$.

**Threshold.** A threshold of 0.01 was used for SPORTS UNDERSTANDING.

## D.2  KNOWLEDGE GENERATION DETAILS

**Post-processing.** In our knowledge generation experiments for both **THINKSUM** and the auxiliary knowledge approach, we post-process the generated knowledge statements, to ensure formatting does not harm the predictions of each method. We first remove the extra spaces and the numbers and punctuation generated by the LLM before each fact while enumerating the items of the list. Later, we only keep sentences that contain only one of the objects of interest from the task, to make sure each sentence contains a knowledge statement into which any of the objects can be substituted. Finally, sentences with less than 3 words are removed as these are not likely to contain informative statements.

**Auxiliary knowledge.** For auxiliary knowledge experiments, we prepend the generated and post-processed knowledge statements before the question in the task. An example is illustrated in Figure D.1.

Table C.3: Few-shot demonstrations used for chain of thought (Table C.2).

| | |
|---|---|
| ODD ONE OUT | Words: blue, pink, magenta, banana
All words are colors except banana. The odd one out is banana.

Words: pencil, eraser, baby, rule, notebook
All words are office supplies except baby. The odd one out is baby. |
| PHRASE RELATEDNESS | For each word or phrase, identify the most related choice from the listed options.
Input: Ice Cream
Option: Antarctica
Option: Titanic
Option: Dessert
Option: Sour Cream
Ice cream is a type of dessert. Therefore, ice cream and dessert are the most related.
Answer: Dessert |
| KNOWN UNKNOWNS | What was the population of San Francisco in 2018?
Option: 879,676
Option: Unknown
The question asks the population of San Francisco in 2018, for which data can be collected. Population data for cities on a yearly basis is available, and thus the answer is known, and it is 879,676.
Answer: 879,676
What was the population of San Francisco yesterday?
Option: 891,402
Option: Unknown
The question asks the population of San Francisco yesterday. As it is not possible to know the exact population of a city on a daily basis, the answer for this question is unknown.
Answer: Unknown |
| LOGICAL DEDUCTION | On a table, there are five plates: a black plate, a white plate, a green plate, a blue plate, and a red plate. The white plate is bigger than the green plate. The red plate is the biggest. The black plate is bigger than the blue plate. The black plate is smaller than the green plate. Which plate is the smallest?
Option: The red plate is the smallest.
Option: The black plate is the smallest.
Option: The white plate is the smallest.
Option: The green plate is the smallest.
Option: The blue plate is the smallest.
The black plate is bigger than the blue plate. The black plate is smaller than the green plate, as a result the green plate is bigger than the blue plate as well. The white plate is bigger than the green plate, which is bigger than the blue plate. As a result, the green plate is bigger than the blue plate. The red plate is the biggest, so it is bigger than the blue plate. Since all other plates are bigger than the blue plate, the blue plate is smallest.
Answer: The blue plate is the smallest. |
| INVENTED WORDS | The word 'borger' are animals who bite specific things for fun, and the word 'folpt' is a type of a chewy toy. Question: Which of the following sentences best characterizes borger folpts?
Option: Borger folpts are leashes for animals.
Option: Borger folpts are toys for infants.
Option: Borger folpts are hard to swallow.
Option: Borger folpts are pet toys.
Borgers are animals, and folpts are chewy toys. Therefore, borger folpts are chewy toys that animals, or pets, can play with. Therefore, the answer is borger folpts are pet toys.
Answer: Borger folpts are pet toys. |

Table D.1: List of examples for the studied BIG-bench tasks.

---

**Task:** PHRASE RELATEDNESS
**Input:** For each word or phrase, identify the most related choice from the listed options.
Input: home town
Option: town center
Option: location
Option: native city
Option: home run

---

**Task:** ODD ONE OUT
**Input:** Pick the odd word out: glass, head, arm, leg, hand, foot

---

**Task:** CONCEPTUAL COMBINATIONS: INVENTED WORDS
**Input:** The word 'binne' means any animal that is furry and has four legs, and the word 'bam' means a simple sort of dwelling. Question: Which of the following sentences best characterizes binne bams?
Option: Binne bams are for pets.
Option: Binne bams are typically furnished with sofas and televisions.
Option: Binne bams are luxurious apartments.
Option: Binne bams are places where people live

---

**Task:** NOVEL CONCEPTS
**Input:** What do the following have in common? 1) bumble bees 2) 01010101 3) race cars
Option: They all make noise.
Option: They all are yellow.
Option: They all are binary.
Option: They all go fast.
Option: They all have stripes.

---

**Task:** LOGICAL DEDUCTION: FIVE OBJECTS
**Input:** On a shelf, there are five books: a gray book, a red book, a purple book, a blue book, and a black book. The red book is to the right of the gray book. The black book is to the left of the blue book. The blue book is to the left of the gray book. The purple book is the second from the right.
Option: The {gray, red, purple, blue, black} book is second from the right.

---

**Task:** CODENAMES
**Input:** Try to identify the 2 words best associated with the word WHITE from the following list: book, anchor, rainbow, shoulder, tunnel, sack, drum, pacific, page, mark, gear, glacier. Give your answer in alphabetical order.

---

**Task:** KNOWN UNKNOWNS
**Input:** How often did Abraham Lincoln cut his toenails?
Option: Every Saturday night
Option: Unknown

---

**Task:** SPORTS UNDERSTANDING
**Input:** Statement: Draymond Green threw a touchdown. Plausible/implausible?

---

**Task:** EMOJI MOVIE
**Input:** What movie does this emoji describe? {lying-face-emoji}
Option: pinocchio
Option: heat
Option: the big sleep
Option: inception
Option: lilo & stitch

---

**Task:** MISCONCEPTIONS RUSSIAN
**Input:** Vaccines cause autism. / Vaccines do not cause autism.  *[in Russian]*

---

