# OpenReview forum: "ThinkSum: Probabilistic reasoning over sets using large language models"
_ICLR.cc/2023/Conference — Submitted to ICLR 2023_

### Official Review · Reviewer_2TCH · 2022-10-23

**Confidence:** 3
**Correctness:** 4
**Technical Novelty And Significance:** 2
**Empirical Novelty And Significance:** 3
**Recommendation:** 8

**Clarity, Quality, Novelty And Reproducibility:**

The paper uses standard benchmark problems and is quite clear in terms of the description and setup and therefore to me it seems like the results should be reproducible,
The novelty may not be as high since it mainly is an integration of methods but the impact of the approach seems significant. The writing is clear, given that they have addressed a large number of different problems with their approach.

**Strength And Weaknesses:**

Strengths
Simple and intuitive idea but produces excellent results in a challenging domain
Seems to be more scalable since it can help us achieve state-of-the-art results with simpler models
The experiments seems quite detailed and impactful since they show in most of the tasks the proposed approach yields state-of-the-art results

Weaknesses
One general issue may be how does the model scale up as the sets produced by the prompts to LLMs become large. In general, probabilistic inference may tend to become less reliable in this case so does that have a major influence on the SUM process.

**Summary Of The Paper:**

The paper describes a 2-stage approach to perform reasoning in large language models (LLMs). The main idea is to decouple the fast thinking process of generating text for a prompt and the slower process of reasoning such as iteratively generating related sequences of text, etc.

The main motivation is to make LLMs more robust to prompt design by decoupling the step to answer prompts and the step to perform the actual reasoning over a complex task. To perform reasoning, the approach proposes to use probabilistic inference. Specifically, GPT-2 XL is used to generate sets of associations and probabilistic inference sums over these associations.

The paper seems to propose a simple idea but one that seems to work very well on a complex suite of tasks in Big-bench. In particular, it is shown that the approach achieves state-of-the-art results on 8 of the ten tasks. For each of the taks, three variants of GPT are used for prompting and the type of probabilistic inference performed is more specific to the task.

**Summary Of The Review:**

Overall seems like an impactful paper with strong empirical results. They show that with their two-step approach, it is possible to achieve state-of-the-art results with much smaller models and thus should have an impact on scalability in future applications.

---

> ### Author Response · Authors · 2022-11-10
> **Authors' response**
>
> Thank you for your constructive comments.
>
> > Specifically, GPT-2 XL is used to generate sets of associations and probabilistic inference sums over these associations.
>
> Small correction: We experiment both with GPT-2 and GPT-3 in this paper, although Figure 1 uses GPT-2 XL for illustration.
>
> > One general issue may be how does the model scale up as the sets produced by the prompts to LLMs become large. In general, probabilistic inference may tend to become less reliable in this case so does that have a major influence on the SUM process.
>
> Please see the response (the first question) to Reviewer KxRo.
>
> > The novelty may not be as high since it mainly is an integration of methods but the impact of the approach seems significant.
>
> Thanks for pointing this out. Our main novelty in this paper lies in a paradigm shift in prompting: we demonstrate that the integration of simple probabilistic inference operations can benefit large language models greatly. We believe that prompt engineering or tricks are not the ultimate solutions to fully developing the potential of LLMs. We appreciate that you see the significant impact of our approach.

---

### Official Review · Reviewer_Xf9p · 2022-10-26

**Confidence:** 3
**Correctness:** 2
**Technical Novelty And Significance:** 1
**Empirical Novelty And Significance:** 1
**Recommendation:** 3

**Clarity, Quality, Novelty And Reproducibility:**

The paper is not well-written and hard to follow; even the definition of the ThinkSum paradigm is not clearly presented.


**Strength And Weaknesses:**

Though the authors present ThinkSum as a general framework for LLM inference in Section 2, the definitions for "Think" and "Sum" are very vague and do not well relate to the example shown in Figure 1. Then the rest of the paper is devoted to explain how ThinkSum can be used to tackle different tasks selected from BIG-bench in a case-by-case sense: each subsection covers one (class of) task, but it remains somewhat unclear how each example relates to the definition of ThinkSum presented in Section 2 and the connection between the individual tasks is unclear. Many new terminologies (in different colors) are introduced: some are completely unnecessary (e.g. product aggregation) and some do not have any clear explanations. The only baseline that the authors compare against to is the few-shot learning with GPT-3; the authors might want to include other prompt-based approaches as baselines.

**Summary Of The Paper:**

This paper proposes ThinkSum: a two-stage paradigm for performing inference with large language models (LLMs) with no gradient update. The authors distinguish ThinkSum from chain-of-though like prompting methods by claiming that ThinkSum performs probabilistic inference instead of using LLMs to directly generate answers. The authors demonstrate the ThinkSum approach on some selected datasets from the BIG-bench and show that it outperforms vanilla few-shot learning with GPT-3.

**Summary Of The Review:**

Despite the extensive demonstrations performed in this study, the authors need to present the ThinkSum paradigm in a more organized and clear way: even the core idea of ThinkSum is not clearly explained the examples throughout the paper do not help much explaining ThinkSum paradigm. The experiments demonstrate the effectiveness of the approach in some sense but need comparisons against more baselines.

---

> ### Author Response · Authors · 2022-11-10
> **Authors' response**
>
> Thank you for your constructive comments.
>
> > Though the authors present ThinkSum as a general framework for LLM inference in Section 2, the definitions for "Think" and "Sum" are very vague and do not well relate to the example shown in Figure 1 [...] it remains somewhat unclear how each example relates to the definition of ThinkSum presented in Section 2 and the connection between the individual tasks is unclear.
>
> This is partly due to the exposition order: We make examples explicit in the Experiments section, which requires undue patience from the reader. To resolve this, we can rename sections and move some parts of the text around for a clear description.
>
> > Many new terminologies (in different colors) are introduced: some are completely unnecessary (e.g. product aggregation) and some do not have any clear explanations.
>
> Product aggregation was used in the *Odd one out*, *Novel concepts*, and *Misconceptions Russian* tasks. (We mistakenly wrote that *Odd one out* used mixture aggregation; this has been fixed in the revised version.)
>
> We would appreciate any specific comments about which Think and Sum techniques have unclear explanations.
>
> > The only baseline that the authors compare against to is the few-shot learning with GPT-3; the authors might want to include other prompt-based approaches as baselines.
> >
> > The experiments demonstrate the effectiveness of the approach in some sense but need comparisons against more baselines.
>
> Please see the response to Reviewer nmJR, where we discuss existing, new, and forthcoming comparisons with prompt-based / chain-of-thought approaches.
>
> > The paper is not well-written and hard to follow; even the definition of the ThinkSum paradigm is not clearly presented.
>
> > Despite the extensive demonstrations performed in this study, the authors need to present the ThinkSum paradigm in a more organized and clear way: even the core idea of ThinkSum is not clearly explained the examples throughout the paper do not help much explaining ThinkSum paradigm.
>
> We have done the following to make the paper organized, explaining the framework of Think and Sum in Section 2 before presenting every single task, using different colors to highlight specific operations belonging to Think and Sum categories; we also have a plan about focusing on explaining three to four tasks and the ThinkSum paradigm in the main text, while moving the rest of the tasks to the appendix. We are wondering whether the reviewer believes this would be a better organization for the paper.

---

### Official Review · Reviewer_KxRo · 2022-10-27

**Confidence:** 2
**Correctness:** 2
**Technical Novelty And Significance:** 2
**Empirical Novelty And Significance:** 2
**Recommendation:** 3

**Clarity, Quality, Novelty And Reproducibility:**

The paper is easy to read. Though, I may have missed some important parts of this paper.

**Strength And Weaknesses:**

The performance of the proposed model is strong in their experiments. However, I am not sure if the improvement can be applied to other tasks. For example, the task in Figure 1 requires evaluating every possible combinations of the choices. This is doable for 5 choices, but not so much for 100 choices or more (in SQuAD-type QA, the output could be any combinations of tokens) which is extremely expensive. Prompting LMs to score all possible candidates is also not accurate if the number of candidates is large, because the union of errors.

The ThinkSum framework, however, is too general such that almost all existing methods fall into this framework, if you think scoring is the Think step, and argmax is the sum step.

Also, tasks experimented in this paper are classification tasks. You may consider running some regression tasks as well.

**Summary Of The Paper:**

The paper proposed a reasoning framework, ThinkSum, which includes a fast think module and a slow sum module. The authors argued that this framework is good at performing many reasoning tasks with simple operations.

**Summary Of The Review:**

The scope of the proposed ThinkSum framework for LLMs is not well defined. Please consider narrowing it down to a subset of tasks. The authors may also consider running some more challenging tasks to show the efficacy of the proposed framework.

---

> ### Author Response · Authors · 2022-11-10
> **Authors' response I: Answering comments and questions**
>
> Thank you for your constructive comments.
>
> > The ThinkSum framework, however, is too general such that almost all existing methods fall into this framework, if you think scoring is the Think step, and argmax is the sum step.
>
> We thank the reviewer for the comment. However, we believe this is not necessarily a downside for our method. Other works, such as chain-of-thought and its extensions, propose specialized prompting tricks and show that they work well on a relatively small set of tasks, yet still require choices by the prompt engineer. Thus, the application of chain-of-thought to a new task often requires a cumbersome prompt engineering procedure, limiting their applicability. We are proposing a more general way of thinking, with an external inference step, that handles a much larger set of tasks. The goal of this generality is to make the adaptation of ThinkSum to new tasks as easy as possible, with few modifications to the Think and Sum steps.
>
> > Prompting LMs to score all possible candidates is also not accurate if the number of candidates is large, because the union of errors.
>
> - The Sum operation is more like averaging than like summing, on the level of output token probabilities. A larger number of candidates is likely to **reduce** the error.
> - As an example, when searching for the most likely "binne bam" in Fig. 1, randomly sampling one possibility (e.g., rabbit) might have high variance, whereas sampling multiple options and averaging the probabilities can serve as a means of variance reduction. The details of this task are given in 3.2.2, where the Sum operation indeed performs averaging of probabilities, which ideally should **reduce** the error (see Fig. 2 (middle), formerly Fig. 3, where accuracy improves with an increased number of candidates).
>
> > Also, tasks experimented in this paper are classification tasks. You may consider running some regression tasks as well.
>
> We address NLP tasks from BIG-bench. We are not aware of regression tasks in that dataset, or standard NLP benchmarks with regression objectives, for that matter. We are wondering what you had in mind.
>
> > The authors may also consider running some more challenging tasks to show the efficacy of the proposed framework.
>
> BIG-bench is a challenging benchmark proposed recently with tasks that are barely solved better than random guessing with direct prompting. The motivation of this benchmark, as stated by its abstract, is "to focus on tasks that are believed to be beyond the capabilities of current language models". Thus, we focus on this benchmark and solve **10** tasks from it. If the reviewer believes there are more challenging benchmarks for current LLMs that we are not aware of, we would be glad to consider them as part of this work, or future work.
>
> > The scope of the proposed ThinkSum framework for LLMs is not well defined. Please consider narrowing it down to a subset of tasks.
>
> If this is the problem (rather than adding more or different tasks), it is easy to fix. In the final version of the paper, we can narrow down the experiments in the main text to 2-3 exemplary tasks (e.g., *Invented words*, *Odd one out*, and *Logical deduction*) and move the rest into the appendix.

---

> ### Author Response · Authors · 2022-11-10
> **Authors' response II: On summing over large sets**
>
> > I am not sure if the improvement can be applied to other tasks. For example, the task in Figure 1 requires evaluating every possible combinations of the choices. This is doable for 5 choices, but not so much for 100 choices or more (in SQuAD-type QA, the output could be any combinations of tokens) which is extremely expensive.
>
> 1. In our experiments on BIG-bench tasks deemed difficult for LLMs, we did not need to sample many possibilities (see the ablation in Fig. 2 (middle), formerly Fig. 3 where performance saturates after 7 examples). Typically, if a task requires more computation than just directly scoring or generating the output tokens, then it can be deemed a hard task, and can't be solved by direct prompting. For example, the *Logical deduction* task with a large number of objects would make ThinkSum expensive, but direct prompting or prompt engineering cannot work at all. Thus, in such cases where prompting approaches cannot be applied, ThinkSum provides an alternative approach, regardless of problem size.  As LLMs become better at hard tasks, the community can always keep inventing harder tasks and applications where the existing approaches need additional computation. ThinkSum provides a methodology for such augmentations of additional probabilistic computations for tasks that are not possible to address with direct prompting.
> 2. If many additional tokens of knowledge do need to be generated to solve a problem, the inference approach we propose can be more efficient than approaches that append this auxiliary knowledge to the prompt (see, e.g., the end of section 3.1.2 and the figure referenced there). Those approaches degrade as the prompt becomes long and incur computation costs that grow cubically with the number of tokens that intervene between the prompt and the answer. On the other hand, generating many facts in parallel and summing over them, as we do, is linear in the number of facts and handles large numbers of facts more gracefully. (Also, as shown in experiments, auxiliary knowledge alternatives underperform ThinkSum.)
> 3. Finally, note that once the computation needed for reasoning is cast as probabilistic inference, then we can rely on a rich body of work on how to speed it up, e.g., sampling methods. A simple way to approximate the exponential evaluation in the example in Fig. 1 (which does not need it) is to run the iterative conditional modes algorithm, where a combination of a binne and a bam is found by keeping one variable fixed (e.g., binne to a particular animal), while the possibilities for bam are evaluated to find the highest-likelihood one; then the bam is fixed and all possibilities for binne are evaluated to find the best matching one, and this is iterated till a best "binne bam" combination is found (probably "dog house"). ICM is linear in both the size of the bam set and the size of the binne set, rather than quadratic. Again, such methods have been developed and studied extensively in statistics and ML communities.

---

### Official Review · Reviewer_nmJR · 2022-10-27

**Confidence:** 3
**Correctness:** 2
**Technical Novelty And Significance:** 2
**Empirical Novelty And Significance:** 2
**Recommendation:** 3

**Clarity, Quality, Novelty And Reproducibility:**

Clarity, Quality and Reproducibility need to be improved.
Without clear writing, it is hard to evaluate novelty.

**Strength And Weaknesses:**

Strengths:
1) Experimental results suggest the proposed method has an improvement over few-shot GPT.

Weaknesses:
1) The paper should reduce the usage of color in the main text
2) The paper should reduce the amount of branding and focus on the actual technique.  I don't see how this has much to do with thinking fast and slow
3) Figure 1 is confusing.  The think step involves manipulating the inputs to get new sequences instead of just generating a set of relevant words?  Then, the second step is the actual summing over the sequences generated in the first step.
4) There is a lack of technical details.  In particular, it is not exactly clear how the sum step is performed.  If this follows a probabilistic framework, please write down the steps formally.
5) This paper first contrasts to chain of thought prompting, but it doesn't compare to that in experiments.
6) Table 1 is confusing.  What's the difference between GPT-3 175B (davinci) and InstructGPT?

**Summary Of The Paper:**

1. This paper proposed a two-stage method that reasons over sets of objects or statements.
2. The paper performs empirical evaluation on the BIG-bench benchmark and shows improvements over fewshot GPT3

**Summary Of The Review:**

The writing is not clear, making it hard to evaluate the contribution of this paper.

---

> ### Author Response · Authors · 2022-11-10
> **Authors' response I: Answering comments and questions**
>
> Thank you for your constructive comments.
>
> > The paper should reduce the amount of branding and focus on the actual technique. I don't see how this has much to do with thinking fast and slow
>
> An example of the relation to fast thinking is the generation of examples or lists through calls to the LLM with simple prompts such as "what is a simple dwelling?" which can be addressed by a fast thinking step to arrive at examples "hut, cabin, cottage" for either a human or an LLM. The analogy to slow thinking arises from making final predictions based on probabilities over sets, which enables our method to evaluate different possibilities or combinations in a "slow" fashion. We believe such analogies to thinking techniques go beyond branding and shed light on reasoning mechanisms in LLMs. Previous studies have also pointed out the benefits of a two-stage approach for reasoning tasks. [1]
>
> [1] Nye, M., Tessler, M., Tenenbaum, J., & Lake, B. M. (2021). Improving coherence and consistency in neural sequence models with dual-system, neuro-symbolic reasoning. Advances in Neural Information Processing Systems, 34, 25192-25204.
>
> > Figure 1 is confusing. The think step involves manipulating the inputs to get new sequences instead of just generating a set of relevant words? Then, the second step is the actual summing over the sequences generated in the first step. There is a lack of technical details. In particular, it is not exactly clear how the sum step is performed. If this follows a probabilistic framework, please write down the steps formally.
>
> The Think step here involves generating a set of relevant words, then manipulating the inputs by substituting the generated words. (The exact prompt used to do this is given in Appendix B.3.) The Sum step involves summing over the scores of the candidate answers under all of the possible substitutions.
>
> In the example in Fig. 1, Sum is referred to as marginalization. The formal probabilistic framework is introduced in Section 3, where probabilities (likelihoods) are computed using calls to the LLM itself. In Fig. 1, the probability of each of the 25 strings such as "A cat hut is a place for animals", "A cat cabin is a place for animals", etc., is computed using parallel calls to LLM. Then, these string probabilities are used to compute the posterior probabilities shown in the figure. The details of the probabilistic framework for the invented words task illustrated in Fig.1. are given in Section 3.2.2. The explanation in Section 3.2.2 can be moved to the Introduction section for clarity if that would be preferable.
>
> > Table 1 is confusing. What's the difference between GPT-3 175B (davinci) and InstructGPT?
>
> These are different variants of the 175B-parameter GPT-3 model. The basic variant (davinci) was trained on a crawl of the Internet. InstructGPT (text-davinci-001) was finetuned on user data to perform better on certain input formats and is often better at in-context learning.

---

> ### Author Response · Authors · 2022-11-10
> **Authors' response II: On chain-of-thought prompting**
>
> > This paper first contrasts to chain of thought prompting, but it doesn't compare to that in experiments.
>
> "Chain of thought" refers to the insertion of generated tokens between the prompt and the (generated or scored) answer. The model is taught, using few-shot demonstrations, how to generate these intermediate tokens. We compare with approaches that add **extracted** (from a knowledge base or an auxiliary LM call), not generated (within the LM's linear workspace) token sequences after the prompt, for the *Odd one out*, *Phrase relatedness*, and *Logical deduction* tasks (see Table B.1 in the Appendix).
>
> Thus, in principle, with suitable examples, it may be possible for a "chain of thought" approach to replace the "Think" phase of ThinkSum. Our auxiliary knowledge baselines make precisely that generous assumption and focus the comparisons on the need for parallel calls and reasoning over possibilities using probabilistic inference (instead of leaving it to the LLM to make the right conclusions from the list of extracted alternatives).
>
> Although we expect that appending facts in a standard format to the prompt would help the model more than teaching the model to generate these facts, we experimented with chain-of-thought approaches and found them to perform far worse, on average. For example, for the *Odd one out* task, a model can be taught to generate the distinguishing property of the "odd" object before generating the answer, as follows:
>
> ```
> Words: blue, pink, magenta, banana
> All words are colors except banana. The odd one out is banana.
>
> Words: pencil, eraser, baby, rule, notebook
> All words are office supplies except baby. The odd one out is baby.
> ```
>
> Although this demonstration does lead the model to generate answers to new problems in the correct **format**, it is still prone to mistakes, e.g.:
>
> ```
> Words: square, circle, triangle, trousers, rectangle, pentagon
> All words are shapes except rectangle. The odd one out is rectangle.
> ```
>
> where the second line was generated by the LM. GPT-3 (davinci) has an accuracy of 35% on the task with this approach, slightly better than direct prompting (27%) but far worse than appending generated facts (from the Think phase) to the prompt (71%) or our approach (84%). These results, and those for several other tasks, will be added to a revision to appear soon.
>
> The main difference between ThinkSum approaches and chain of thought or other auxiliary knowledge approaches is that the format is **not constrained to reason within the linear structure of text**.

---

### Author Response · Authors · 2022-11-10
**To all reviewers: Updated version posted**

We thank all reviewers for their feedback. We would like to alert you that we just uploaded a first revision of the paper, making minor changes to fix errors and improve the exposition.

---

### Author Response · Authors · 2022-11-18
**Update: Chain of thought results**

Dear reviewers,

We have just updated the paper with comparisons to chain-of-thought prompting on five tasks and two GPT-3 model variants (see new Table C.2 for the results, which we also reproduce below, and Table C.3 for the prompts and demonstrations used). As expected, these approaches usually improve over direct prompting, but are universally outperfomed by ThinkSum.

| Model | davinci | davinci | davinci | text-davinci-002 | text-davinci-002 |
|:---|:---|:---|:---|:---|:---|
| **Task** | **Direct** | **CoT** | **ThinkSum** | **CoT** | **ThinkSum** |
| *Odd one out*  | 0.27 | 0.33 | 0.80 | 0.64 | 0.84 |
| *Phrase relatedness* | 0.58 | 0.55 | 0.85 | 0.79 | 0.87 |
| *Logical deduction* | 0.32 | 0.25 | -- | 0.39 | 0.77 |
| *Known unknowns* | 0.61 | 0.70 | 0.54 | 0.74 | 0.76 |
| *Invented words* | 0.29 | 0.50 | 0.64 | 0.64 | 0.71 |

As we discuss in the new Appendix C.1 and in the response to Reviewer nmJR (part II), this supports our hypothesis about the importance of doing inference over probabilities obtained by parallel calls to the LLM.

Please let us know if you have any questions.

---

### Decision · Program_Chairs · 2023-01-20

**Decision:**

Reject

**Justification For Why Not Higher Score:**

Lack of significance

**Justification For Why Not Lower Score:**

N/A

**Metareview: Summary, Strengths And Weaknesses:**

The paper proposes a two-step technique that achieves some empirical improvement on bigbench.
The extensive philosophical discussions around thinking fast and slow are not appropriate for an AI research paper, and what is fundamentally a very ad-hoc trick.
Overall it's an interesting find, but not significant enough for a research publication.